# Lasso-grafting of macrocyclic peptide pharmacophores yields multi-functional proteins

Emiko Mihara [1], Satoshi Watanabe[1], Nasir K. Bashiruddin[2], Nozomi Nakamura[1], Kyoko Matoba[1], Yumi Sano [1], Rumit Maini[2], Yizhen Yin[2], Katsuya Sakai [3,4], Takao Arimori [1], Kunio Matsumoto [3,4], Hiroaki Suga [2✉] & Junichi Takagi [1✉]

Protein engineering has great potential for devising multifunctional recombinant proteins to serve as next-generation protein therapeutics, but it often requires drastic modifications of the parental protein scaffolds e.g., additional domains at the N/C-terminus or replacement of a domain by another. A discovery platform system, called RaPID (Random non-standard Peptides Integrated Discovery) system, has enabled rapid discovery of small de novo macrocyclic peptides that bind a target protein with high binding specificity and affinity. Capitalizing on the optimized binding properties of the RaPID-derived peptides, here we show that RaPID-derived pharmacophore sequences can be readily implanted into surface-exposed loops on recombinant proteins and maintain both the parental peptide binding function(s) and the host protein function. We refer to this protein engineering method as lasso-grafting and demonstrate that it can endow specific binding capacity toward various receptors into a diverse set of scaffolds that includes IgG, serum albumin, and even capsid proteins of adeno-associated virus, enabling us to rapidly formulate and produce bi-, tri-, and even tetra-specific binder molecules.

[1] Laboratory of Protein Synthesis and Expression, Institute for Protein Research, Osaka University, Osaka, Japan. [2] Department of Chemistry, Graduate School of Science, The University of Tokyo, Tokyo, Japan. [3] Division of Tumor Dynamics and Regulation, Cancer Research Institute, Kanazawa University, Kanazawa, Japan. [4] WPI-Nano Life Science Institute (WPI-NanoLSI), Kanazawa University, Kanazawa, Japan. ✉email: hsuga@chem.s.u-tokyo.ac.jp; takagi@protein.osaka-u.ac.jp

Macrocyclic peptides represent a class of synthetic compounds that is rapidly gaining attention as a new drug modality, particularly in the therapeutic intervention of protein–protein interfaces (PPIs)[1,2]. Compared to traditional small molecule drugs, their footprint on the binding surface of the target protein is generally comparable to that of typical antibody. Thus, macrocyclic peptides have high specificity and affinity despite their small to medium size (MW in the range of 1000–3000 Da). By using the Random non-standard Peptides Integrated Discovery (RaPID) system that combines mRNA-display with genetic code reprogramming, we have reported many binders for various targets by isolating de novo macrocyclic peptides from a pool of random sequences consisting of more than $10^{12}$ members (Fig. 1a)[3,4]. Such peptides generally exhibit exquisite binding specificity and high affinity toward the target and have proven to be highly useful in applications such as receptor antagonists or agonists, crystallization chaperones, and imaging/detection tools[5–9]. Despite the fact that this method offers extraordinary speed for identifying potent species, the resulting macrocyclic peptides are not necessarily ready for drug use because they often suffer from unpredictable pharmacokinetics and bioavailability[10]. The ability to first rapidly identify

pharmacophores from macrocyclic peptides and then modularly integrate them into natural human proteins would produce a more reliable drug format that bears the merits of both macrocycles and proteins.

In this study, we show that the binding ability of the RaPID-derived macrocyclic peptides can be maintained when the thioether ring-closure moiety is replaced by a well-folded natural protein domain. Serendipitously, this does not demand any special protein engineering efforts and can be achieved by simply taking the internal sequence of a RaPID peptide and simply inserting this lasso-like moiety into a surface-exposed loop from a wide range of proteins. We refer to this protein engineering method as lasso-grafting and demonstrate that it can endow specific binding capacity toward various receptors into a diverse set scaffolds that includes IgG and serum albumin. Lasso-grafting enabled us to rapidly formulate and produce bi-, tri-, and even tetra-specific binder molecules. Moreover, this extraordinarily facile method enabled us to generate a target-specific AAV vector that can infect cells solely via the lasso-peptide–receptor interaction.

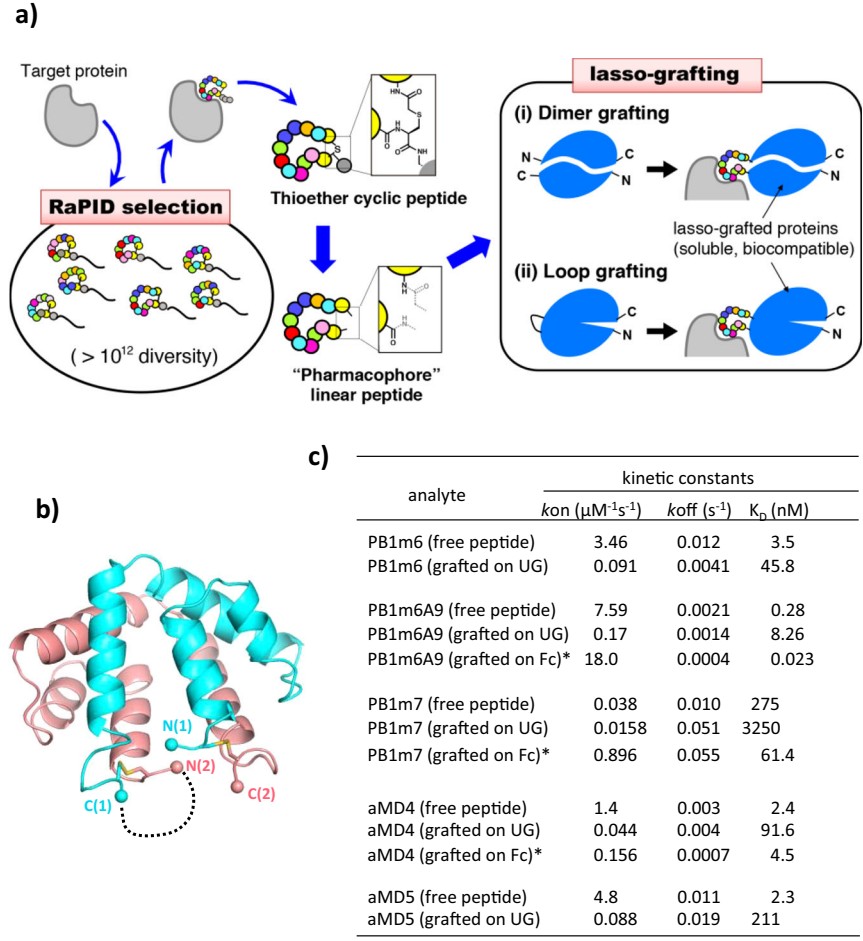

**a**

**c**

| analyte | kinetic constants | | |
|---|---|---|---|
| | $k$on ($\mu M^{-1}s^{-1}$) | $k$off ($s^{-1}$) | $K_D$ (nM) |
| PB1m6 (free peptide) | 3.46 | 0.012 | 3.5 |
| PB1m6 (grafted on UG) | 0.091 | 0.0041 | 45.8 |
| | | | |
| PB1m6A9 (free peptide) | 7.59 | 0.0021 | 0.28 |
| PB1m6A9 (grafted on UG) | 0.17 | 0.0014 | 8.26 |
| PB1m6A9 (grafted on Fc)* | 18.0 | 0.0004 | 0.023 |
| | | | |
| PB1m7 (free peptide) | 0.038 | 0.010 | 275 |
| PB1m7 (grafted on UG) | 0.0158 | 0.051 | 3250 |
| PB1m7 (grafted on Fc)* | 0.896 | 0.055 | 61.4 |
| | | | |
| aMD4 (free peptide) | 1.4 | 0.003 | 2.4 |
| aMD4 (grafted on UG) | 0.044 | 0.004 | 91.6 |
| aMD4 (grafted on Fc)* | 0.156 | 0.0007 | 4.5 |
| | | | |
| aMD5 (free peptide) | 4.8 | 0.011 | 2.3 |
| aMD5 (grafted on UG) | 0.088 | 0.019 | 211 |

**b**

**Fig. 1 Conversion of RaPID-derived cyclic peptides into protein grafts. a** Flow-chart of the lasso-graft method. Once a high affinity cyclic peptide is obtained against a target via the RaPID system in the discovery phase (left), the sequence information is used to formulate binding proteins in the grafting phase (right) by inserting the peptide either (i) in between the N- and C-termini of a dimeric protein or (ii) in the middle of an exposed loop of monomeric protein. **b** Disulfide-stabilized homodimeric structure of uteroglobin (PDB ID: 2UTG). The dotted line shows the peptide-grafting site between the C-terminus of molecule 1 (cyan) and the N-terminus of molecule 2 (salmon). **c** Kinetic binding constants for RaPID-derived peptides before and after the lasso-grafting. Binding kinetics of UG or Fc proteins lasso-grafted with either PlxnB1-binding peptides (PB1m6, PB1m6A9, and PB1m7) or MET-binding peptides (aMD4 and aMD5) was determined for their respective target molecules using SPR (actual sensorgrams are shown in Fig. S1c–h). Values for free peptides are taken from Matsunaga et al.[6] (for PB1m6 and PB1m7), Bashiruddin et al.[14] (for PB1m6A9), and Ito et al.[5] (for aMD4 and aMD5). Kinetic parameters for the grafted Fc proteins (*) do not represent their true 1:1 affinity due to the divalent nature.

## Results

**Grafting of RaPID peptides into a uteroglobin scaffold.** The RaPID selection methodology is a powerful platform that derives target-specific high-affinity macrocyclic peptides from a vast pool of candidate peptides. Although all library peptides must be cyclized by a thioether bond during selection[11], this ring-closure may be replaced by other chemical structure with equivalent geometries. This is possible because the target-binding is generally mediated by binding-competent amino acid residues from the middle of the loop (Fig. 1a, left) rather than residues near the cyclized termini. Most importantly, our available X-ray structures from peptide–protein co-crystals indicated that the peptides adopt a well-folded conformation, and we observed a wide range of tertiary structures over the represented library sequences[12]. We hypothesized that this peptide pharmacophore structure can be maintained if the close proximity of the N- and C-termini on the peptide is reinforced by fusion to a protein domain (Fig. 1a, right (i)). We first searched for a dimeric protein suitable for this fusion topology and found uteroglobin (UG). UG is a small secreted homodimeric protein comprised of two ~70 residue chains. In the UG crystal structure, the N-terminus of each chain is proximal to the C-terminus of the opposite chain, and the conformation of the termini is stabilized by two nearby disulfide-bridges (Fig. 1b)[13]. To test the hypothesis that the core sequence of a RaPID peptide can be displayed on a protein of choice and still retain its binding function, we tested constructs based on a linearized version of the 17-residue PB1m6 macrocyclic peptide (referred to as m6 hereafter) that is known to bind with human Plexin B1 (PlxnB1) with a high affinity[6]. We generated an expression plasmid containing m6 fused between two UG monomers (referred to as UG$_2$-m6). This method is referred to as the "dimer grafting" scheme of lasso-grafting hereafter (Fig. 1a, right (i)). This single-chain tandem UG dimer (Fig. S1a) expressed well in mammalian cells and purified from the culture supernatant when fused to a C-terminal (His)$_6$ tag (Fig. S1b). A surface plasmon resonance (SPR) experiment showed that this UG$_2$-m6 retained the binding activity of parental m6 peptide toward PlxnB1 (Figs. 1c and S1c). Although the binding affinity of the UG$_2$-m6 was ~13-fold lower than that of the original cyclic peptide (Fig. 1c), this was mainly due to the slower on rate ($k_{on}$), which can be explained in part considering the reduction in the diffusion rate caused by the increase in the molecular size (2031 Da for free peptide vs 18,506 Da for UG$_2$-fusion protein). As the $k_{off}$ value remained unchanged (in fact, the grafted version showed 3-fold slower rate), the overall stability of the formed complex is comparable between UG$_2$-cyclized m6 and thioether-cyclized m6. We also tested that alternative lasso sequences can convey binding upon grafting using an affinity-matured variant of m6. This matured m6A9 sequence was recently isolated by screening after partially randomizing the internal sequence of m6[14]. When m6A9 was grafted into the UG tandem dimer to make UG$_2$-m6A9, this construct also showed specific binding toward PlxnB1 with a similar $k_{off}$ to that of the free peptide (Figs. 1c and S1d). We next applied the same UG$_2$-fusion strategy to different kinds of RaPID peptides including another PlxnB1-binding peptide (PB1m7)[6] and two MET-binding peptides (aMD4 and aMD5)[5] (Table S1). All these UG$_2$-peptide fusions exhibited specific binding toward their respective targets with $k_{off}$ values comparable to those of the free peptides (Fig. 1c and Fig. S1f–h), confirming that the grafting compatibility is not a special property of PB1m6/m6A9 but is a shared property of many RaPID macrocyclic peptides, despite completely different sequences and target-dependencies. Interestingly, a UG$_2$-m6 mutant with the ring-closing disulfide-Cys residues mutated to Ser showed binding kinetics very similar to that of the parental fusion (Fig. S1d, e). We conclude that the disulfide tether connecting the loop termini was not essential for binding in the context of the UG fusion construct.

**Lasso-grafting is applicable to various protein scaffolds.** The above observations motivated us to build another hypothesis. As the grafted-peptide portion displayed on UG$_2$ (Fig. 1b, dotted line) is topologically equivalent to a loop protruding from a globular protein, we anticipated that ring-closure could be achieved by simply grafting the peptide pharmacophore into an open-cut point in the middle of a loop in an arbitrary protein. This method is referred to as the "loop grafting" scheme of lasso-grafting hereafter (Fig. 1a, right (ii)). In order to test this hypothesis, we chose 4 β-sandwich fold domains as scaffolds for grafting: the 10th FN-III module of human fibronectin (Fn10), the first IgV domain of human carcinoembryonic antigen (CEA), the first IgV domain of signal regulatory protein alpha (SIRPα), and an anti-GFP single-domain antibody (VHH). The internal peptide sequence of m6A9 or aMD4 were grafted into 2 sites (s1 or s2) both located at the tip of β-hairpin loops (Fig. 2a–d and Table S2), based on the assumption that such a topology would be ideal to maintain functional conformation of the parental macrocyclic peptides. When expressed as fusion proteins with a C-terminal Fc-tag, these peptide-grafted β-sandwich domains were secreted well and showed specific binding to their respective targets, PlxnB1 (for m6A9) and MET (for aMD4), in an immunoprecipitation-like pulldown assay (Fig. 3a–d). These results confirm that the lasso-grafting is compatible with various β-sandwich domains. We also found that as many as eight locations on the β-hairpin-rich Fc domain could accommodate peptide lasso-grafting (Fig. 2e), and all lasso-grafted Fc fusions were binding competent for the target of the parent RaPID peptide (Fig. 3e). Furthermore, the lasso-grafted Fc fusions were able to bind to their target receptors expressed on cell surface (Fig. 4a, b). The lasso-grafted Fc scaffold is particularly useful because it is compatibile with common reagents like Protein A and anti-Fc secondary antibodies and because it is intrinsically divalent in nature. Since Fc is a homodimer of the CH2-CH3 domains of the IgG heavy chain, the lasso-grafted version can bear two identical peptide moieties on one molecule. This divalent nature is highly desirable, because the grafted Fc molecules show very high apparent binding affinity toward their targets due to the avidity effect (Fig. 1c). We next examined if protein domains other than the β-sandwich fold are compatible with this strategy. We performed lasso-grafting of m6A9 and aMD4 peptides at selected loop sites of four protein scaffolds that belong to three different classes: all-α (human serum albumin, HSA, and human growth hormone, hGH), β-barrel (retinol-binding protein, RBP), and α/β (alkaline phosphatase, ALP) fold topologies, resulting in a total of 18 constructs (Fig. 2f–i). In the pulldown assay, lasso-grafting resulted in the desired near-stoichiometric target binding for 16 out of the 18 constructs (Fig. 3f–i). We conclude that the core pharmacophore of high-affinity RaPID peptides can be grafted successfully into various stably folded protein domains because they were selected for binding while their termini were constrained to close proximity. To further evaluate if the method is robust and effective with multiple peptide-target pairs, we lasso-grafted the s8 site of the Fc protein with each of three distinctive peptide pharmacophores from macrocycles isolated in our laboratory for binding to EGFR (A6-2f), TrkB (tkD5), or α6β1 integrin (IB8; see Table S1). FACS analysis of the lasso-grafted Fcs revealed that they specifically bind to cells expressing their respective target receptor proteins (Fig. 4c–e). From these results, we conclude that the RaPID peptides for many different targets can be readily grafted into various protein scaffolds.

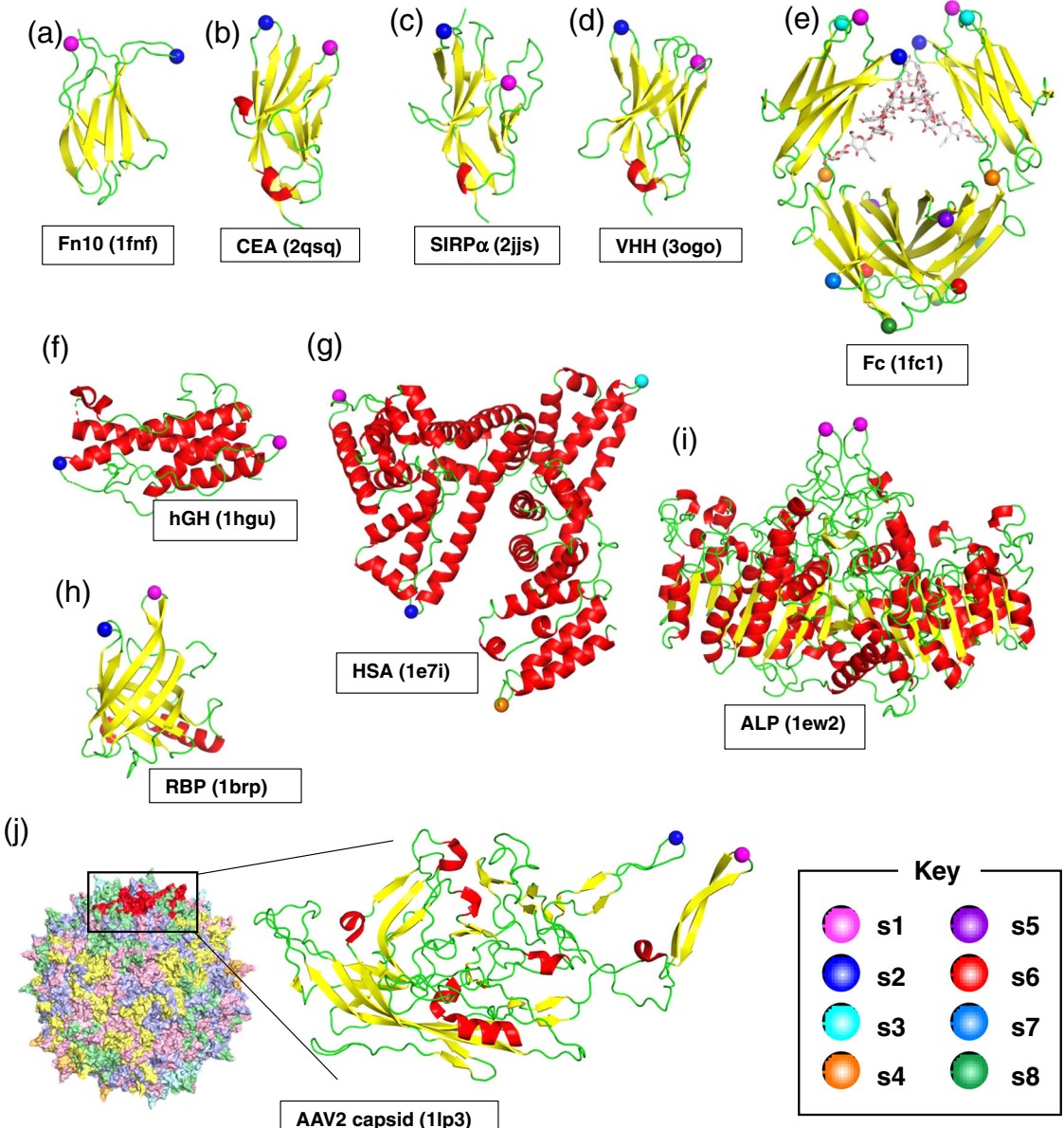

**Fig. 2 Location of the peptide pharmacophore grafting sites.** The structure of protein scaffold used for lasso-grafting are shown in cartoon representation: **a** fibronectin 10th type-III domain (Fn10, PDB ID: 1FNF), **b** carcinoembryonic antigen (CEA, PDB ID: 2QSQ) 1st IgV domain, **c** signal regulatory protein alpha (SIRPα, PDB ID: 2JJS) 1st IgV domain, **d** anti-GFP single-domain antibody (VHH, PDB ID: 3OGO), **e** IgG1 Fc (PDB ID: 1FC1), **f** human growth hormone (hGH, PDB ID: 1HGU), **g** serum albumin (HSA, PDB ID: 1E7I), **h** retinol-binding protein (RBP, PDB ID: 1BRP), **i** placental alkaline phosphatase (ALP, PDB ID: 1EW2), and **j** VP3 capsid protein from AAV serotype 2 (PDB ID: 1LP3). In **j**, structure of the whole virus-like particle comprising 60 VP3 subunits is also shown, with each subunit colored uniquely. For all panels, grafting sites (s1 up to s8) are indicated by spheres with distinct colors as shown in the key. The actual amino acid sequences flanking each grafting site is shown in Table S1. Note that two equivalent sites are present for each site for the homodimers, Fc and ALP.

**Instant creation of multi-specific binders by lasso-grafting.** The above proof-of-concept studies using the Fc domain stimulated us to apply the method of lasso-graft in the format of more medically applicable protein scaffolds. We first applied it in generating multi-specific antibodies, a prominent biological drug modality. Since any IgG scaffold consists of a common Fc domain and two Fab domains with highly variable antigen-binding sites, we envisioned that lasso-grafted RaPID peptides in the Fc domain of an IgG would readily add another binding function without disturbing the bivalency of the Fab's antigen-binding site. To test this idea, we lasso-grafted three representative RaPID peptides (m6A9, aMD4, and A6-2f) to the Fc domain of three medically

relevant antibodies, anti-neuropilin-1(Nrp1) YW64.3[15], anti-PD-L1 avelumab[16], and anti-CD3 OKT3[17]. FACS analysis revealed that each antibody retained its parental antigen-binding capacity for receptors expressed on cells (Fig. 5a–c, top-row panels), indicating that lasso-grafting in the Fc domains did not disrupt the function of the Fabs. Critically, lasso-grafting a peptide pharmacophore to the Fc domain of a functional antibody successfully generated a bi-specific antibody (Fig. 5a–c). In addition, sequential lasso-grafting of multiple peptides onto unique sites on the anti-Nrp1 Fc resulted in an corresponding sequential gain of binding functions, ultimately enabling creation of an antibody with $1 + 3$ binding specificity (Fig. 5a, the far-right

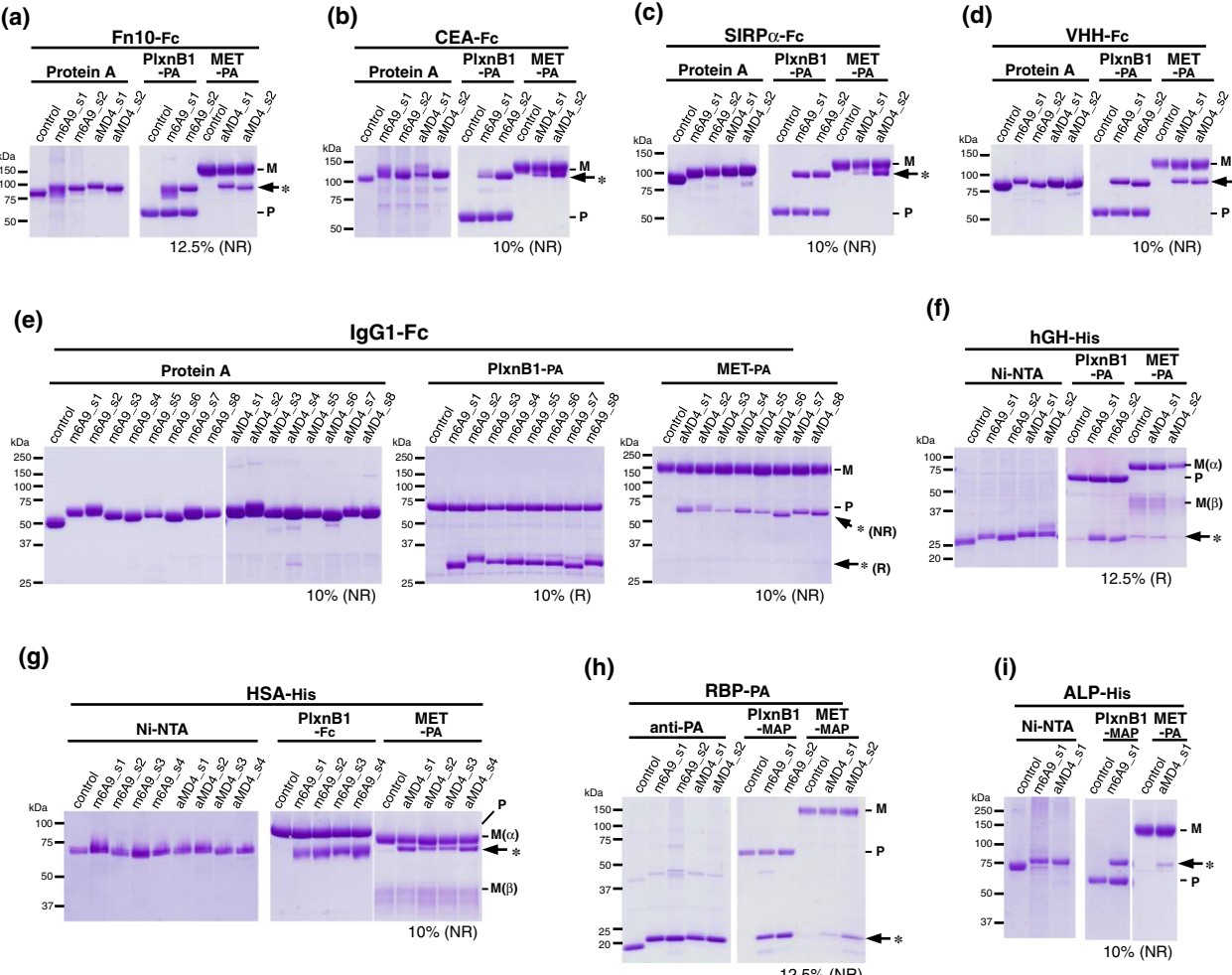

**Fig. 3 Lasso-grafting of binder peptides into various protein scaffolds.** Nine different protein scaffolds including Fn10-Fc (**a**), CEA-Fc (**b**), SIRPα-Fc (**c**), VHH-Fc (**d**), IgG1 Fc (**e**), hGH (**f**), HSA (**g**), RBP (**h**), and ALP (**i**) were grafted with m6A9 (PlxnB1 binder) or aMD4 (MET binder) peptides at different sites. In the labels, each graft design is denoted as *peptide name_insertion site*, e.g., m6A9_s4. Expression and secretion of these grafts and non-grafted protein controls, were checked by pulldown experiments (left half of each panel) using beads conjugated with Protein A (for Fc-containing proteins, **a**–**e**), Ni-NTA (for His-tagged proteins, **f**, **g**, **i**), or the anti-PA tag antibody NZ-1 (for PA-tagged proteins, **h**). The same set of samples were subjected to pulldown using PlxnB1- or MET-bound beads (right half of each panel), see Methods section for details on preparing beads bound via affinity tag (including PA, MAP, and Fc) to PlxnB1 and MET ectodomain fragments. Coomassie-stained SDS-PAGE gels loaded with protein eluted from beads and run under non-reducing (NR) conditions, except for those gels run under reducing (R) conditions to avoid overlapping bands between the grafted proteins and MET or Plxn. The acrylamide concentration in the gel and the running conditions (NR or R) are indicated below each gel image, and migration positions for MET(M), PlxnB1 (P), and the grafted proteins (*) are shown on the right. Data are representative of at least two independent experiments. Uncropped gel images are provided in the Source Data file.

panels labeled "triple"). We refer to this multi-specific antibody format as an addbody hereafter and note that there are at least eight grafting-compatible sites on the Fc portion (Fig. 2e) of an addbody. Further, the peptide pharmacophore grafted into the Fc is on the opposite end of the addbody molecule from the original antigen-binding site in two Fab domain. This structure opens the possibility for simultaneously engaging two antigens on opposing cell surfaces. To test the utility of our addbody format in such applications, the OKT3-based addbody grafted with the MET-binding aMD4 peptide (see Fig. 5c) was incubated with MET-expressing CHO cells together with CD3-expressing Jurkat cells. The aMD4-bearing OKT3 addbody clearly induced heterotypic cell–cell attachment while the parent OKT3 antibody induced no attachment above background (Fig. 5d), indicating that this addbody could efficiently bridge two specific cell types in a manner similar to the bispecific antibodies used in the therapies for redirecting immune effector cells to tumor cells[18].

**Lasso-grafting into AAV capsid**. We next took advantage of the modularity of the lasso-grafting method to change the cellular tropism of adeno-associated virus (AAV). One of the most well-studied and promising gene-delivery vehicles, AAV, is a small, naked icosahedral virus with 60-subunit capsid composed solely of the product of the *Cap* gene[19]. AAV harbors binding sites toward various cellular receptors, such as proteoglycans and AAVR[20]. Because these receptors are expressed on wide range of cells, AAV has an intrinsically broad tropism and lacks the specificity needed to deliver genes to a particular tissue(s) or organ (s). While some success in targeting AAV to specific tissues has been achieved with capsid engineering by mutagenesis-based random screening and/or insertion of non-viral protein moieties into the capsid[21–24] it is prohibitively labor-intensive to impart new specificity on AAV. Therefore, we predicted that lasso-grafting would enable us to readily alter the specificity of AAV in a modular fashion using pre-optimized peptide pharmacophores

from RaPID. To test this approach, we attempted to first disrupt the normal infectious ability of AAV by grafting a peptide with no cell surface receptor engagement into a loop in the Cap capsid protein and then restore function to the AAV-peptide fusion

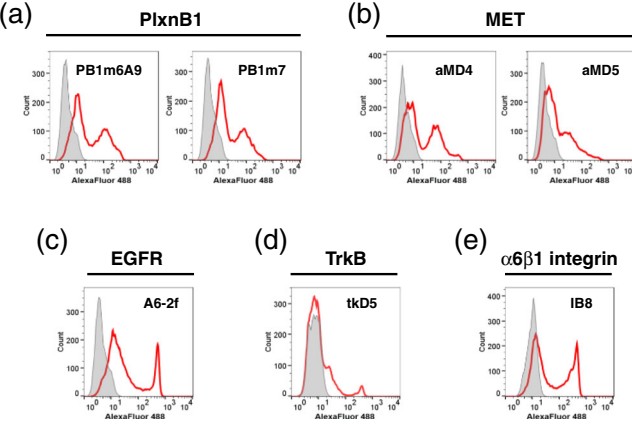

**Fig. 4 FACS analysis of peptide-grafted Fc protein binding to cell surface target receptors.** HEK293T cells transiently expressing PlxnB1 (**a**), MET (**b**), EGFR (**c**), TrkB (**d**), or α6β1 integrin (**e**) are incubated with each peptide-grafted Fc (red histogram) or control Fc (gray histogram), followed by labeling with Alexa Fluor 488-labeled anti-human Fc.

capsid by grafting in a new receptor-specific peptide pharmacore at the same graft site. To first identify an exogenous peptide-grafting site that would disrupt gene transduction but not prevent viral packaging, we grafted a 12-residue peptide tag with no receptor engagement (PA tag,[25]) into the *Cap* gene of AAV serotype 2 (AAV2) at each of two loops (s1 and s2, see Fig. 2j and Table S2) that had been reported to tolerate a peptide insertion[26]. As expected, both mutant AAV2s (PA_s1-Cap$^{AAV2}$ and PA_s2-Cap$^{AAV2}$) were efficiently produced in HEK cells at a titer comparable to that of the wild-type virus (Fig. 6a), indicating successful viral assembly and packaging. In contrast, the gene transduction activity was affected as desired by one of the mutations; while the s1 mutant retained activity comparable to that of wild-type AAV, the s2 mutant completely lost its capacity to infect (Fig. 6a). This deleterious effect from the s2 modification on AAV2 infectivity is likely due to the disruption of binding to AAV2-specific receptor heparan sulfate proteoglycan[24] and/or the pan-AAV endocytosis receptor AAVR[27]. By using this PA_s2-Cap$^{AAV2}$ incapable of infecting cells as a capsid scaffold, we tested if presentation of m6A9 peptide grafted on the capsid could mediate gene delivery into PlxnB1-expressing cells. Unfortunately, when the capsid was solely composed of the m6A9_s2-Cap$^{AAV2}$, the virus showed very low titers (generally less than 3% of the wild-type virus), suggesting that it is not fully compatible with capsid assembly and/or gene packaging. We therefore decided to employ a chimeric virus format, where only a

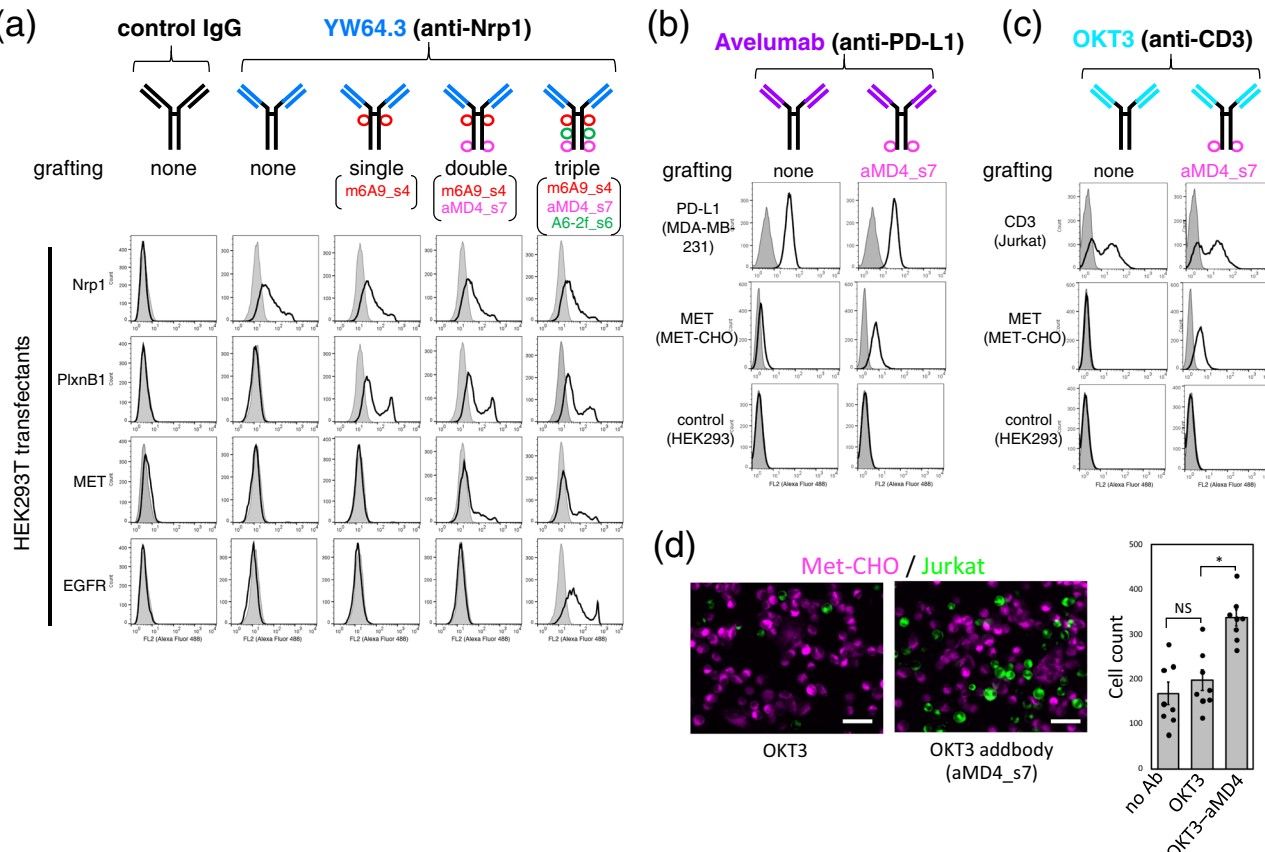

**Fig. 5 Instant formulation of multi-specific antibodies by peptide grafting onto IgG, addbodies. a–c** Flow cytometric analysis of grafted antibody binding to cells expressing antigens (indicated on the left). Antibody scaffolds include anti-neuropilin-1 YW64.3 (**a**), anti-PD-L1 avelumab (**b**), and anti-CD3 OKT3 (**c**). Each antibody design is depicted schematically above the panels. The gray histograms in **a** represent binding to untransfected HEK293T cells, while those in **b** and **c** are for the labeling of the same cells with the IgG control. **d** Addbody-mediated heterotypic cell engagement. DiI-labeled MET-CHO cells (magenta) adhered onto the plate were overlaid with DiO-labeled Jurkat cells (green) and incubated with OKT3 (left) or the OKT3-aMD4 addbody (right) for 30 min. After washing, the remaining cells were photographed (bar, 100 µm). The number of attached Jurkat cells was counted for multiple fields ($n = 8$) and shown in the right as mean ± SD (*$p = 0.0021$, two-sided *t*-test; NS not significant). Source data are provided as a Source Data file.

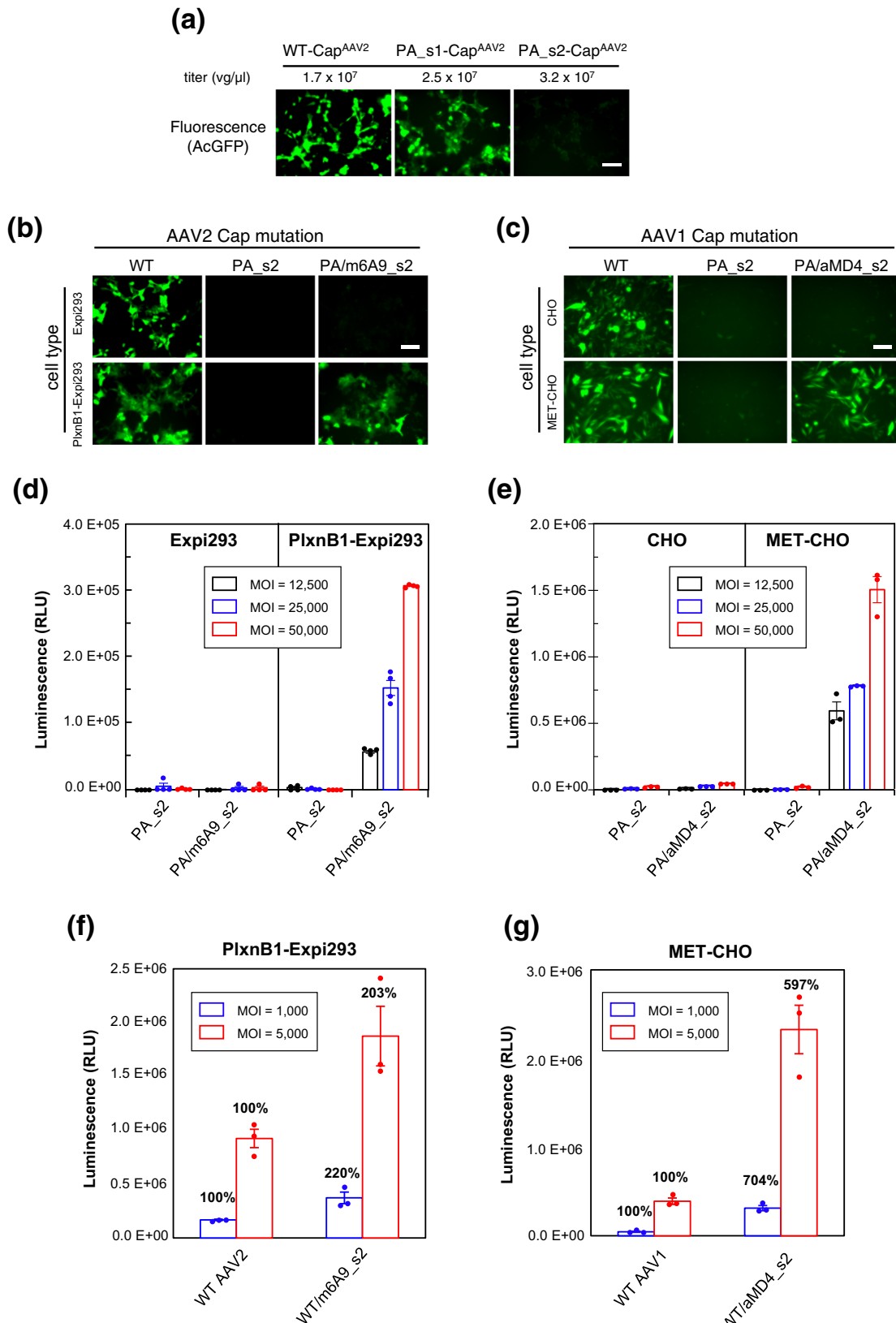

fraction of capsid subunits are substituted with m6A9_s2-Cap$^{AAV2}$. By co-transfecting cells with plasmids coding for PA_s2-Cap$^{AAV2}$ and m6A9_s2-Cap$^{AAV2}$ at a DNA ratio of 9:1, a chimeric virus incorporating a fraction of m6A9_s2-Cap$^{AAV2}$ subunits per capsid was prepared. The resultant chimeric virus successfully transduced the AcGFP gene into Expi293F cells that

stably expressed human PlxnB1 (Fig. 6b), while no detectable AcGFP expression was observed with the parent cell line after incubation with the chimeric virus. We next expanded this analysis to another AAV serotype (AAV1) using MET-binding aMD4 peptide as the targeting moiety. Unmodified AAV1 capsid was capable of transducing both Chinese hamster ovary (CHO)

**Fig. 6 Peptide grafting onto AAV capsids enables acquisition of different cellular tropism. a** Effect on capsid function from PA tag insertion into AAV2 Cap VP3 at loops s1 or s2. AcGFP-containing AAV2 with either wild-type (WT) or mutant capsids (PA_s1 and PA_s2) were recombinantly produced in AAVpro 293T cells and their titers were measured by qPCR (indicated above each panel). The virus stocks were then used to infect adherent Expi293F cells at MOI = $6 \times 10^5$ vg/cell. Fluorescent micrographs are shown for the cells after 48 h of infection. Scale bar, 100 μm. **b**, **c** Receptor-dependent gene transduction with mutant AAV2 (**b**) or AAV1 (**c**). For PlxnB1-dependent transduction, cells stably expressing PlxnB1 or the parent cells were infected with wild-type (WT) or one of the two mutant AAV2 particles containing an AcGFP gene at an MOI of $6 \times 10^5$ vg/cell and then imaged after 24 h. For MET-dependent transduction, unmodified CHO cells or MET-overexpressing CHO cells were infected with WT or mutant AAV1 particles at an MOI of $5 \times 10^4$ vg/cell, and photographed after 48 h. The PA_s2 mutant virus contains PA tag at the S2 site on all capsid subunits, while the chimeras (PA/m6A9_s2 or PA/aMD4_s2) contain PlxnB1-binding m6A9 peptide or MET-binding aMD4 peptide at the same site in a fraction of capsid subunits. Scale bar, 100 μm. **d**, **e** Transduction with AAV-Luc. Indicated cells were infected with AAV2 (**d**) or AAV1 (**e**) capsid mutants carrying a luciferase reporter gene at varying MOI values. The cells were then analyzed for the luciferase activity after 48 h. **f**, **g** Enhancement of the gene transduction efficiency of the WT AAVs by additional presentation of receptor-specific peptides. PlxnB1-expressing Expi293F cells (**f**) or MET-expressing CHO cells (**g**) were infected with either WT viruses or chimeric viruses bearing m6A9 or aMD4 peptides at the s2 site at MOI of 1000 (blue bars) or 5000 (red bars) and measured for the luciferase activity. The relative transduction efficiency of each mutant virus compared to the WT virus in each condition is shown above the bar. Data are reported as the mean ± SD of $n = 4$ (for **d**) and $n = 3$ (for **e–g**) technical replicates from a representative experiment out of 9 (**d**) or 2 (**e–g**) independent ones. Source data are provided as a Source Data file.

cells and MET-overexpressing CHO cells (MET-CHO), while the engineered AAV1 virus composed of mutant PA_s2-Cap$^{AAV1}$ showed minimum transduction with both cell lines as desired (Fig. 6c). In contrast, the chimeric AAV1 composed of PA_s2-Cap$^{AAV1}$ with a fraction of aMD4_s2-Cap$^{AAV1}$ subunits was able to transduce the AcGFP gene in a manner completely dependent on the presence of MET (Fig. 6c). To more quantitatively assess the gene transduction capability of these receptor-oriented chimeric viruses, we evaluated the transduction of a luciferase reporter gene. As shown in Fig. 6d, e, dose-dependent luciferase induction in cells expressing PlxnB1 and MET was observed with both m6A9-grafted and aMD4-grafted chimeric viruses, respectively. Most importantly, these viruses could not transduce parental cells that do not express the target receptor. This result confirms that an interaction between the target receptor and the grafted peptide is a strict requirement for infection.

**Lasso-grafted AAV can transduce cells at lower dose**. The above approach was powerful for granting exclusive target-receptor specificity to the virus, but the overall transduction efficiency of the engineered viruses was lower than that of the parent virus, because the peptide grafting disrupts the natural attachment/entrance machineries. This is not necessarily desirable for engineering viruses to overcome the serious drawback of high dosing requirements with current gene therapy. Therefore, we next tested if the lasso-grafting could produce viruses with enhanced transduction efficiency compared to that of the wild-type AAVs. In experiments similar to those described in the previous section, mutant Cap subunits grafted with PlxnB1-binding m6A9 or MET-binding aMD4 peptides were co-expressed with the corresponding WT Cap2 or Cap1 subunits during viral production in HEK293 cells. As the majority of the Cap subunits are intact in these viruses, these chimeras showed sufficient transduction of parental cells at MOI = 5000, which was 96% (for Expi293F cells) and 44% (for CHO-K1 cells) of 100% WT viruses. When they were used to transduce cells that express their corresponding target receptors, they exhibited 2-fold (for m6A9-grafted AAV2, Fig. 6f) or >6-fold (for aMD4-grafted AAV1, Fig. 6g) increase in the transduction efficiency over the WT viruses. Of note, roughly the same level of luciferase expression was achieved by WT AAV1 at MOI = 5000 and by the aMD4-chimera at MOI = 1000, indicating that the capsid engineering succeeded in reducing the amount of virus required in this model system by a factor of 5.

## Discussion

Various technology platforms for obtaining artificial antibodies or antibody-like binder molecules for specific targets have been developed that emulate the ability of the natural immune system to produce an almost infinite number of antibodies against yet-to-be encountered foreign molecules. One platform that exemplifies the application of this concept is the broadly used phage-display method[28]. Although great conceptual and experimental variations exist among these technologies, they generally start with a library of proteins whose surface-exposed loop(s) are randomized, followed by panning with an immobilized target molecule to select for variants with binding specificity. Arguably, these library-based selection methods represent the most successful examples in modern biotechnology and have led to the discovery of numerous useful biomolecules. However, the first-generation binders obtained from such partially randomized protein libraries often show only a modest level of affinity due to limited library diversity. Thus, it is often necessary to conduct additional selection campaigns to identify variants with improved binding that are worth practical application[29]. This second step usually involves directed evolution or saturation mutagenesis[30,31], and leads to a significant extension of development time and increase the uncertainty of the success.

As with the conventional methods described above, lasso-graft generates proteins with de novo loop sequences. However, the path to achieve this goal is markedly different; we separate the discovery step involving selection and optimization of loop sequence via RaPID system from the grafting step that simply involves insertion of the optimized loop sequences into structurally permissive sites in a variety of protein scaffolds. In this way, one does not have to deal with difficulties in constructing library of randomized loop(s) in a specific protein for each experiment; rather, one can take full advantage of the RaPID system that utilizes a vast library of short (<20-residue) cyclic peptides with $10^{12}$ members, leading to rapid discovery (generally within a few weeks) of highly optimized binder peptides against target proteins at a remarkably high success rate. The next challenge arises if the biologically active peptide should be integrated into a protein scaffold. Generally, recombinant fusion is limited to the termini. Grafting into an independently folded domain is typically a risky approach due to the possibility for disturbing the local structure. However, for peptides whose active structure (i.e., target-bound state) is a lasso-like conformation, loop-insertion often allows for a functionally active internal fusion[25,32]. In fact, similar "binding-domain insertions" has been recently found in several natural antibodies in which target-binding is mediated by the insertion of a small (~30 residues) to large (~100 residues) domain at the tip of the CDR3 loop[33,34].

One of the great advantages of lasso-grafting over existing grafting methods is the wide choice of scaffold proteins and grafting sites. As it is known that heteromeric protein–protein

interactions are often mediated by loops[35,36], a loop-grafting can be a promising way to transfer PPI onto unrelated proteins. However, only a limited number of special protein scaffolds have reported to present PPI-mediating loop peptides in a binding-competent conformation, and each scaffold has limited grafting sites[37–39]. Choosing an appropriate loop peptide motif is also challenging, because the conformation of a flexible peptide loop is difficult to predict in the context of the new scaffold, and they frequently exhibit low affinity due to the entropic penalty upon binding[36]. We do not know exactly why the RaPID peptide pharmacophores show such high compatibility with the lasso-graft method, but we suspect that the very high library diversity ($>10^{12}$) enabled selection of very rare binders that can adopt a stable and binding-competent conformation with a cyclic scaffold. If the RaPID peptides can autonomously fold into a binding-competent conformation, the same conformation may be achieved by replacing the cyclizing residues with two closely apposed residues in the context of folded protein domain, regardless of the structure of the scaffold and possibly even the grafting site.

Once a target-binding peptide sequence derived from a RaPID screening is confirmed to be compatible with lasso-grafting onto one site on a scaffold, the same peptide pharmacophore can be readily re-grafted to a desired location on the same or a different scaffold (Fig. 3). The usefulness of this feature is best exemplified by its application to the AAV capsid proteins. Unlike soluble globular proteins that exist in isolation (e.g., IgG and HSA), Cap proteins must maintain correct multi-subunit assembly to maintain their native structure. This requirement generally precludes straightforward protein fusion unless the number and/or the sites of the fusion moiety are carefully controlled[23]. However, we believe that the lasso-grafting approach introduces relatively small structural disturbance when applied to a protruding surface-exposed loop. This limited perturbation contributed to the successful grafting of binding-competent PlxnB1- and MET-binding peptides onto AAV2 and AAV1 particles. Importantly, the receptor-targeting moiety does not have to be presented in all 60 subunits in the capsid, and chimeric virus containing relatively small number of peptide-bearing Cap subunits per particle was capable of transducing cells in a manner completely dependent on the receptor expression. It should be noted, however, that the chimeric virus used here could be highly heterogeneous population of mixed subunit compositions, because capsid assembly is a stochastic process. Although AAV can infect and transduce a broad range of cells/tissues through ubiquitously expressed sulfated glycans as well as the prerequisite endocytosis receptor AAVR, there is a need for the development of vectors that can selectively transduce specific types of tissues[26]. The present data suggest that applying the lasso-grafting approach to multiple gene-delivery vehicles including, but not limited to, AAV may enable the use of entry receptors freely chosen from a large pool of cell surface molecules. Lasso-grafting may therefore allow the creation of custom-designed vectors for gene therapy.

Another key feature of the lasso-graft approach is the combinatorial capability, which is particularly important for the development of bi/multi-specific antibodies. Bispecific antibodies are the focus of intensive research due to their potential to target two disease-causing molecules at a time and/or to draw two antigens into close proximity[40]. In addition to a conventional bispecific design of heterodimerized two half IgGs, numerous bi- and multi-specific antibody design modalities exist[18]. However, they tend to have issues with developability, immunogenicity, and manufacturability because of their highly engineered nature. In contrast, the addbody format we report here has only minimal modifications and maintains the global structure of the original IgG. This format enabled the instant development of a tetra-specific binder molecule by simply pasting three peptide grafts onto an existing antibody without applying any special protein engineering know-how (Fig. 5a). Likely, this is possible because we identified eight redundant grafting-compatible sites on the Fc domain, and the Fc scaffold may support even higher combinations of binding functions.

Although we found that the RaPID peptide grafts can be moved to alternate grafting sites throughout a protein with a surprisingly high success rate, there were cases where grafted proteins could not be expressed. We expect that this is failure of the resultant protein to fold, which is clearly dependent on the nature of both the peptide and the grafting site. We envision that a systematic survey of the folding behavior of variously grafted proteins may be useful in studying folding mechanism of the protein during the biosynthesis to deepen our understanding of the protein architecture. Considering the wide selection of viable scaffold proteins, we believe that the lasso-grafting approach can both contribute to an expansion in the scope of protein therapeutics and also find utility in basic research that advances our protein design capabilities.

## Methods

**Selection of the RaPID peptides.** $N$-chloroacetyl-L- or D-Tyr-tRNA[fMet] was prepared by the use of the respective amino acids esterified with cyanomethyl group (L- or D-ClAc-Tyr-CME) incubated with a flexizyme, eFx[41]. Ribosomal synthesis of the macrocyclic peptide library from NNK RNA templates was performed as previously described[42]. In brief, 1.2 μM puromycin-linked mRNA library was translated in a methionine deficient FIT reaction containing 25 μM either L- or D- ClAc-Tyr-tRNA[fMet] for 30 min at 37 °C. The reaction was incubated at 25 °C for 12 min before disruption of the ribosome-mRNA complex by incubation at 37 °C for 30 min in the presence of 20 mM EDTA. The resulting peptide-linked mRNAs were then reverse transcribed using RNase H- reverse transcriptase (Promega) for 1 h at 42 °C and buffer was exchanged for 50 mM Tris-HCl (pH 7.4), 150 mM NaCl, 0.05 vol% Tween-20. Affinity screening was performed by three serial passages (counter selections, 10 min each at 4 °C) of the library over Covalt-NTA or Streptavidin Dynabeads (Life Technologies), followed by affinity selection against 200 nM protein target immobilized on the same beads for 30 min at 4 °C. cDNA was eluted from the beads by heating to 95 °C for 5 min, and fractional recovery from the final counter selection (negative control) and the affinity selection step were assessed by quantitative PCR using Sybr Green I on a LightCycler thermal cycler (Roche). Enriched DNA libraries were recovered by PCR and used as input for transcription reactions to generate the mRNA library for the subsequent round of selection.

For high-throughput sequencing, DNA samples from the final round of selection were amplified, purified using a Nucleospin column (Machery-Nagel) and sequenced using a MiSeq high-throughput sequencer (Illumina). Data analysis was performed using CLC sequence viewer 7 software (Qiagen). The target proteins used to derive peptides described in this paper include biotinylated human PlxnB1 ectodomain[6], human MET ectodomain-Fc[5], biotinylated human EGFR ectodomain[43], His-tagged human TrkB ectodomain, and biotinylated human α6β1 integrin ectodomain[44].

**Cell lines.** Cell lines used in this study were obtained from ATCC (HEK293T, MDA-MB-231), RIKEN BRC Cell Bank (Jurkat), Thermo Fisher (Expi293F), and TAKARA bio (AAVpro 293T). Expi293F cells stably expressing human PlxnB1[6] and CHO-K1 cells stably expressing human MET[45] were established previously. All cell lines were routinely tested for the presence of mycoplasma.

**Protein design, construction, and expression.** To design peptide-grafted proteins, each scaffold protein was structurally inspected to find appropriate grafting site where the Cα distance between the two anchorpoint residues located at chain ends (for dimer grafting, Fig. 1ai) or within an exposed loop (for loop grafting, Fig. 1aii) were <7 Å. Internal sequence of the RaPID-derived cyclic peptides (Table S1) were then inserted into these identified sites (Table S2) with up to three spacer residues of either Gly or Ser at both ends. All expression constructs were made using pcDNA3.1-based backbone with appropriate signal peptide and a tag or fusion partner to allow efficient purification and beads-pulldown assay. For the construction of the single-chain UG linked by the peptide (generally called UG₂-(peptide name)), two DNA fragments coding for human UG (UniProt P11684) and a peptide-coding region were assembled as shown in Fig. S1a by extension PCR. For the construction of Fc-only protein, human IgG1 Fc (residues 104-330, UniProt P01857) was used without any tags. For Fc-fused β-sandwich domains, following regions were amplified from the original cDNAs or synthesized DNAs and fused with the human IgG1 Fc; residues 1539–1631 of human fibronectin (UniProt P02751-15); residues 35–142 of human CEA (UniProt P06731), residue 31–148 of human SIRPα (UniProt P78324), and residue 2–116 of anti-GFP single-domain

antibody (PDB ID: 3OGO,[46]). Full length HSA (UniProt P02768), hGH (UniProt P01241), RBP (UniProt P02753), and ALP (UniProt P05187) coding regions were cloned in the pcDNA3.1 vectors with C-terminal His tag (HSA, hGH and ALP) or C-terminal PA tag[47](for RBP). Using these constructs as templates, peptide-inserted variants were prepared by extension PCR, followed by the verification of DNA sequences. For the construction of full-length IgG containing peptide grafts at the Fc domain (i.e., addbody), the variable regions of heavy and light chains of YW64.3, avelumab, and OKT3 were gene-synthesized using the publicly available amino acid sequences and formulated in the form of human IgG1/kappa, and the peptide-grafting was performed as described above. For the addbodies containing more than one peptide grafting, the extension PCR process was repeated to incorporate different peptide sequences at different sites. The coding regions of all expression constructs were verified by DNA sequencing. A complete list of primers used in the extention PCR to construct peptide grafts is provided as Supplementary Table S3. Protein expressions were performed using Expi293 expression system (Thermo Fisher) unless otherwise indicated. $UG_2$ proteins were purified from the culture supernatants using Ni-NTA-agarose resin as shown in Fig. S1b, buffer-exchanged to 20 mM Tris, 150 mM NaCl, pH 7.5 (TBS), concentrated to ~1 mg/ml, and stored at $-80\,°C$ until used.

**Beads pulldown**. In order to assess the binding ability of various proteins grafted with m6A9 (PlxnB1 binder) or aMD4 (MET binder) in parallel, simple beads-pulldown method was utilized. To this end, soluble ectodomain fragments of human PlxnB1 (residues 1-535) and human MET (residues 1-931) with different tags were expressed and captured onto the beads immobilized with Protein A (for Fc-tagged version), anti-PA tag antibody NZ-1 (for PA-tagged version), or anti-MAP tag[32] antibody PMab-1 (for MAP-tagged version). After brief washing, the beads were further incubated with the culture supernatants containing various peptide-grafted proteins. Bound proteins were then eluted by adding SDS-containing buffer, and analyzed by SDS-PAGE. Binding specificity was confirmed by the lack of nonspecific binding of control scaffold proteins with no peptide grafting.

**Kinetic binding measurement using Biacore (SPR)**. The ectodomain fragments of human PlxnB1(1-535) or human MET(1-931) were C-terminally fused with a biotin acceptor sequence (SSLRQILDSQKMEWRSNAGG) and co-expressed with a biotin ligase (BirA) in Expi293F cells to achieve BirA-mediated biosynthetic biotinylation[6], and immobilized onto a Series S sensor chip SA (GE Healthcare) at a surface density of ~930 RU (PlxnB1) and ~840 RU (MET), respectively. The binding was evaluated by injecting peptide-grafted UG solutions serially diluted using the running buffer (20 mM HEPES-NaOH (pH 7.5), 150 mM NaCl, 0.05% Surfactant P20). The runs were conducted in a single cycle kinetics mode employing the following parameters; flow rate of 30 μl/min, contact time of 120 s, and dissociation time of 300 s. After each run, the surface was regenerated by injecting the regeneration buffer (10 mM Glycine-HCl (pH 3.0), 1 M NaCl) until the response returned to the original baseline level. The binding curves of the measurement cell (immobilized with PlxnB1/MET) were subtracted with that of reference cell (unimmobilized), and used to derive kinetic binding values. Data were obtained using a Biacore T200 instrument (GE Healthcare) at 25 °C, and the results were analyzed by using Biacore T200 evaluation software version 3.0.

**Flow cytometry**. To measure binding of peptide-grafted Fc proteins to the respective target receptors, HEK 293T cells were transiently transfected with plasmids coding for various full-length human receptors including PlxnB1, MET, EGFR, TrkB, integrin α6β1, or Nrp1 using X-tremeGENE HP (Merck #6366236001), and detached from dishes by a brief treatment with trypsin/EDTA at 2 days post transfection, followed by an incubation with peptide-grafted Fc proteins diluted at ~10 μg/ml for 1.5 h. After washing twice with PBS, cells were incubated with Alexa Fluor 488-labeled goat anti-human IgG (1:400 dilution, Thermo Fisher, A11013) at room temperature for 30 min. To measure binding of peptide-grafted IgG (addbodies), either the HEK293T transient transfectants or cell lines with endogenous expression of PD-L1 (MDA-MB-231), MET (MET-CHO), and CD3 (Jurkat) were used. Stained cells were analyzed on an EC800 system (Sony) and the data were analyzed with FlowJo software (Tomy Digital Biology).

**Heterotypic cell–cell attachment assay**. The CHO cells stably expressing human MET[45] were labeled by a fluorescent membrane marker DiI (Biotium, #30023) and plated in a 24-well plate (Thermo Fisher, #142475) at $5 \times 10^5$ cells/well in F12 growth medium containing 10% FCS. After 5 h, the cells were overlaid with Jurkat cells labeled by fluorescent membrane marker NeuroDiO (Biotium, #30021), which had been preincubated with purified OKT3 or aMD4-grafted OKT3 addbody at 5 μg/ml for 1.5 h on ice, and then further incubated for 30 min at 37 °C. After removing the non-adherent Jurkat cells by gentle washing with ice-cold PBS for three times, the samples were fixed with 4% paraformaldehyde in PBS for 30 min at room temperature, and the phase-contrast and fluorescence cell images were recorded using TRITC filter set (for MET-CHO) and GFP filter set (for Jurkat) for at least 8 field views with a BZ-X700 digital fluorescence microscope (Keyence). The number of CHO cells (red fluorescence) and Jurkat cells (green fluorescence) were analyzed by using the hybrid cell count software (Keyence), and the average

number of Jurkat cells attached per area were evaluated as the score of addbody-mediated heterotypic cell-cell attachment.

**Recombinant production of AAV mutants and gene transduction assays**. Recombinant AAV2 vector was prepared using an AAVpro Helper Free System (TAKARA, #6230). As the source of wild-type and mutant Cap subunits to prepare peptide-grafted AAV2 capsid, we modified the pRC2-mi342 plasmid encoding *Rep* and *Cap* genes. In order to avoid unwanted mutations introduced in this large (8.2 kb) plasmid, we first introduced unique AgeI and NheI sites flanking s1 and s2 sites to prepare pRC2-mi342AN using the primers described in the Supplementary Table S4. Various grafted-peptide mutants were made on the ~0.8-kb AgeI-NheI fragment in a separate vector, and swapped into pRC2-mi342AN after sequence verification. As a result, pRC2-mi342AN plasmids with the following three mutations were constructed; PA_s1, PA_s2, and m6A9_s2. For the production of recombinant AAV1 vectors, pAAV2-1 plasmid carrying *Rep* gene from AAV2 and *Cap* gene from AAV1 (Addgene, #112862) was used. As in the case of AAV2 construction, mutations (PA_s2 and aMD4_s2) were introduced in the ~0.8-kb BsiWI-SbfI fragment in a separate vector before the final cloning in the pAAV2-1 plasmid. AAVpro-293T cell line (AAV-293, TAKARA, and Z2273N) was used for AAV production and was cultured in 10% FCS/DMEM supplemented with 1 % non-essential amino acids (NEAA, Sigma, M7145-100mL) and 0.5% penicillin/streptomycin (P/S, Sigma, P4458-100ML). AAV-293 cells were seeded at $8.0 \times 10^5$ cells/well in six-well cell culture plate (Thermo Fischer, and about 30 min before transfection, the culture medium was changed with 5% FCS/DMEM containing 1% NEAA and 0.5% P/S. Cells were transfected with 1 μg/well of a mixture of modified pRC2-mi342 or pAAV2-1 plasmids, 1 μg/well of pHelper plasmid, and 1 μg/well of pAAV-CMV-Fluc or pAAV-CAG-AcGFP plasmid (a kind gift from Takahisa Furukawa, Osaka University). Transfection was performed using 6 μg of PEI-Max (Polysciences, #24765-1). After 24 h post transfection, the culture medium was changed with 1% FCS/DMEM supplemented with 1% Glutamax (Gibco, #35050-061), 1% NEAA, and 0.5% P/S. Seventy two hours after transfection, cells were collected and recombinant AAV was extracted using AAVpro extraction solution (TAKARA, #6235), and concentrated by ultrafiltration in PBS using Amicon ultra (Millipore, 100 kDa cutoff, UFC510024) when necessary. Virus titers were determined by qPCR analysis of AAV genome copies using AAVpro titration kit ver. 2 (TAKARA, #6233). For the AcGFP gene transduction, Expi293F cells stably expressing human PlxnB1[6] or the parental cells were seeded into each well of 96-well black wall plate (Greiner, #655090) at 5000 cells/well and were infected with WT or mutant AAV2 at MOI = $6 \times 10^5$. Cells were cultured in 5% FCS/DMEM including 1% NEAA, and 0.5% P/S. In gene transduction with WT or mutant AAV1, CHO-K1 cells stably expressing human MET[45] or the parental cells were used and cultured in 5% FCS/Ham's F12 (Wako, 087-08335) containing 0.5% P/S. These cells were seeded at 10,000 cells/well and were infected at MOI = $5 \times 10^4$. Fluorescent microscope images were recorded after two days of infection by BZ-X700 microscope. For the luciferase gene transduction, cells were plated at 10,000 cells/well into 96-well black wall plate and cultured for 1 day, followed by infection with mutant AAV1s and AAV2s at various MOI values. Cells were cultured for additional 2 days in 5% FCS/DMEM containing 1% NEAA and 0.5% P/S or 5% FCS/Ham's F12 containing 0.5% P/S, and the luciferase activity was determined by measuring luminescence for 0.5 s using Luciferase Assay System (Promega, E1501) in a Glomax NAVIGATOR (Promega, GM2000). Three or four technical replicates for each infection condition were measured.

**Reporting summary**. Further information on research design is available in the Nature Research Reporting Summary linked to this article.

## Data availability

All the data supporting the findings of this study are available within the article and its Supplementary Information files, and from the corresponding authors upon reasonable request. Source data are provided with this paper.

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

## Acknowledgements

We would like to thank Keiko Tamura-Kawakami for construction of the expression vectors, and Samuel Thompson for critical reading of the manuscript. This work was supported in part by Japan Agency for Medical Research and Development (AMED), Platform Project for Supporting Drug Discovery and Life Science Research (Basis for Supporting Innovative Drug Discovery and Life Science Research) under JP19am0101090 and 19am0101075 to H.S. and J.T., respectively, Project for Cancer Research and Therapeutic Evolution (P-CREATE) from AMED to Ku.M., and by MEXT KAKENHI Grant Numbers 18K19298 from the Ministry of Education, Culture, Sports, Science and Technology of Japan (MEXT) to J.T.

## Author contributions

E.M. performed and analyzed all the experiments with the exception of the following. S.W. and Y.S. performed and analyzed AAV experiments. N.K.B., R.M., Y.Y., and K.S. performed RaPID peptide selection against various receptors. N.N., Ky.M., and T.A. performed and analyzed UG-fusion experiments and Biacore experiments. Ku.M. supervised experiments on the MET-binder grafts and wrote the manuscript. H.S. and J.T. conceived the experimental design, analyzed the data, and wrote the manuscript. All authors contributed to the preparation of the manuscript.

## Competing interests

H.S. and J.T. are co-founders and shareholders of MiraBiologics Inc. E.M., K.S., and Ku. M. are also shareholders of the same company. All other authors declare no competing interests.
