## [Peer Review File · Nature Communications]

REVIEWER COMMENTS

Reviewer #1 (Remarks to the Author):

This is a very interesting paper that definitely has the potential to change thinking in the field. The authors propose an extension of the already highly successful RaPID technology to use the output cyclic peptides as a pharmacophore that can be adapted to be grafted onto a variety of larger protein constructs. Figure 1 of the paper nicely captures the concept. The approach is applicable not only to conventional proteins but also to dimeric constructs and I think it is the latter that offers greatest potential to even more innovations in the future. For example, I can imagine a range of applications that might involve the incorporation of novel design functions into proteins- e.g. designer hinges to modulate folding/unfolding.

The work appears to have been done to a very high technical standard and involves a range of multidisciplinary analyses to support the interpretations. This would be one of the rare examples of manuscripts I see that have the potential for acceptance more or less without further experimentation. The manuscript could be slightly improved by further proofreading to remove some grammatical or phraseology errors e.g. in the first line of the abstract (...is a promising approach) but this is something that could be attended to during the production process I imagine. My only other suggestion is that the authors might wish to add a few sentences to speculate on broader possible applications of the technology beyond the pharma space. Overall, this is an outstanding piece of work

Minor points

Line 115 when they refer to a disulfide knot, I think disulfide tether would be a more appropriate word- I don't see a knot there?

Throughout- there are minor typos/grammar corrections and that need attention- in supplementary data as well (eg patterns misspelt)

References 6, 15, 16: DOI looks odd

Correct reference 9: Should be - RSC Chem. Biol., 2020, 1, 26-34

Reference 12 needs page number.

Reviewer #2 (Remarks to the Author):

The study by Mihara et al is focused on grafting macrocyclic peptides onto different proteins for improving function. The mRNA display system (RaPID) was developed earlier and this approach is referred to as Lasso grafting. In the case of antibodies, the authors are able to demonstrate new protein-protein interface interactions, while still maintaining the Fab interaction with the antigen. In another example, the study showcases that upto three distinct lassos in the Fc region that expands the range of interactions. It is unclear whether Fc interactions and circulation half-life are affected. These applications are interesting and could potentially impact the monoclonal Ab field provided some significant utility can be demonstrated. However, the study falls short of demonstrating the translatability of this approach. The authors also attempt to graft lasso peptides onto AAV capsid proteins, which appears to be fraught with problems. A major concern is that the overall titers of the AAV2 mutants is exceedingly low, bringing to question whether this approach can be scaled or translated. Viral transduction data using the peptide insertion mutants is primarily in vitro and not quantitative. For the chimeric VP3 insertion mutants, it is unclear how the authors determine the percentages of the different capsid components with peptide inserts in the chimeras. Capsid assembly is a stochastic process and it is likely that a largely heterogeneous population was generated. The ratios of plasmids utilized are unlikely to be actually reflected in particle composition. This is a significant flaw with the scientific premise of this arm of the study. Overall, this is an incremental study primarily evaluating a peptide grafting approach with the exception that the inserts are macrocyclic in nature. Technical aspects and applications of the approach appear to be cumbersome and not readily apparent, respectively.

Reviewer #3 (Remarks to the Author):

Review of NCOMMS-20-39422 "Lasso-grafting of macrocyclic peptide pharmacophores yields multi-functional proteins"

Macrocyclic peptides are a class of synthetic compounds that are gaining increasing attention as possible therapeutic modalities, especially given their ability to target and inhibit protein-protein interactions. Despite their low molecular weights, macrocyclic peptides typically possess larger binding interfaces than traditional small molecule-based drugs, which allows for greater specificity and affinity. Furthermore, easy and rapid methods for generating macrocyclic peptides using strategies such as RaPID have emerged to make these compounds more widely and readily accessible. However, macrocyclic peptides often possess unfavorable pharmacokinetic properties that can limit their therapeutic potential. To address this issue, the authors of this paper utilize two interesting engineering strategies in which they replace the thioether ring-closure moiety of the macrocyclic peptide with a self-folding protein domain, so called "lasso grafting". In essence, they graft the pharmacophore of the macrocyclic peptide onto another protein or proteins, taking advantage of the grafted onto proteins' favorable pharmacokinetic properties while trying to maintain the binding properties of the macrocyclic peptide. In their first strategy, they take advantage of protein dimers in which the C- and N- termini of the individual proteins within the dimer are structurally held in proximity. By fusing the individual proteins into a single polypeptide sequence with the residues of the macrocyclic peptide inserted between the C- and N- termini of the parental proteins, they were able to effectively produce grafted protein dimers. They were able to utilize this strategy to graft multiple different macrocyclic peptide pharmacophores onto a uteroglobin dimer. Interestingly, while the k_{off} s of these grafted proteins were mostly unchanged relative to the parental macrocyclic peptide, the k_{on} s were much lower. As such, the K_D s for each grafted protein was in some cases substantially lower than the corresponding macrocyclic peptide. The authors' second and perhaps more powerful strategy involved grafting the pharmacophore of a macrocyclic peptide into the exposed loops of a protein. Utilizing this strategy, the authors were able to graft pharmacophores of various macrocyclic peptides easily onto many different protein scaffolds, including medically relevant antibodies. Furthermore, for proteins that possess many exposed loops, the authors were able to show that multiple pharmacophores could be grafted onto a single scaffold (ex: one in each exposed loop of an Fc domain). As such, the authors were able to show that they could produce bi-, tri-, and even tetra-specific protein binders. In an interesting application of this engineering strategy, the authors were able to graft various targeting pharmacophores onto capsid proteins of AAV particles in order to rapidly alter virus tropism. In summation, this paper presents an interesting and potentially powerful and rapid engineering strategy to 1) improve the pharmacokinetic properties of macrocyclic peptides and 2) generate multi-specific protein binders.

I have only one comment to address prior to publication. Although the binding kinetics were reported for the first engineering strategy (grafting pharmacophores onto protein dimers), none were reported for the second engineering strategy (grafting pharmacophores into protein loops). It would be important to know how the binding kinetics of some of these loop-grafted proteins correlate to the parental macrocyclic peptides. It is interesting to know if the decrease in K_D s reported for the dimer-grafted proteins could be rescued or even improved in the loop-grafted proteins if multiple loops within a single protein were replaced with the same target specific pharmacophore.

Response to the reviewers' comments

RE: NCOMMS-20-39422

Mihara et al.

" Lasso-grafting of macrocyclic peptide pharmacophores yields multi-functional proteins

"

Followings are our point-by-point responses to the comments and requests provided in the decision letter we received on Nov 17. Requests/comments are in *black and italicized* and our responses are in **red**.

Comments by the reviewers

Reviewer #1:

Remarks to the Author:

This is a very interesting paper that definitely has the potential to change thinking in the field.

The authors propose an extension of the already highly successful RaPID technology to use the output cyclic peptides as a pharmacophore that can be adapted to be grafted onto a variety of larger protein constructs.

Figure 1 of the paper nicely captures the concept. The approach is applicable not only to conventional proteins but also to dimeric constructs and I think it is the latter that offers greatest potential to even more innovations in the future. For example, I can imagine a range of applications that might involve the incorporation of novel design functions into proteins- e.g. designer hinges to modulate folding/unfolding.

The work appears to have been done to a very high technical standard and involves a range of multidisciplinary analyses to support the interpretations. This would be one of the rare examples of manuscripts I see that have the potential for acceptance more or less without further experimentation.

We thank the reviewer very much for the kinds words and for appreciating the value of our work. It is possible that grafting a loop peptide moiety in a protein can change the overall folding/unfolding behavior, and we would like to explore more in this research direction in the future.

The manuscript could be slightly improved by further proofreading to remove some grammatical or phraseology errors e.g. in the first line of the abstract (...is a promising approach) but this is something that could be attended to during the production process I imagine. My only other suggestion is that the authors might wish to add a few sentences to speculate on broader possible applications of the technology beyond the pharma space. Overall, this is an outstanding piece of work

The revised manuscript has now been proofread by a native English speaker doing active research in protein engineering. We also thank the reviewer for encouraging us to speculate on the applications beyond the pharma-related field, which we were reluctant to include in the old MS. We have added a paragraph at the end of the discussion (page 10, line 352-361) that indicates the possible use of the lasso-graft method in basic protein folding studies.

Minor points

Line 115 when they refer to a disulfide knot, I think disulfide tether would be a more appropriate word- I don't see a knot there?

We changed "knot" to "tether" (page 4, line 121). Thank you.

Throughout- there are minor typos/grammar corrections and that need attention- in supplementary data as well (eg patterns misspelt)

The entire manuscript has been proofread by a native English speaker. We believe that all typos and errors have been addressed.

References 6, 15, 16: DOI looks odd

We corrected these references. Thank you.

Correct reference 9: Should be - RSC Chem. Biol., 2020, 1, 26-34

Yes indeed. It has been corrected.

Reference 12 needs page number.

It does not have page numbers because it is an online-only article.

Reviewer #2:

Remarks to the Author:

The study by Mihara et al is focused on grafting macrocyclic peptides onto different proteins for improving function. The mRNA display system (RaPID was developed earlier and this approach is referred to as Lasso grafting. In the case of antibodies, the authors are able to demonstrate new protein-protein interface interactions, while still maintaining the Fab interaction with the antigen. In another example, the study showcases that up to three distinct lassos in the Fc region that expands the range of interactions. It is unclear whether Fc interactions and circulation half-life are affected. These applications are interesting and could potentially impact the monoclonal Ab field provided some significant utility can be demonstrated.

We thank the reviewer for the careful reading and pointing out the potential value of this technology in the field of antibody engineering. We are currently assessing the *in vivo* half-life of the grafted Fc proteins, and the results look very promising (i.e., the lasso-grafted Fc proteins tested so far have exhibited plasma half-lives in mice that are indistinguishable from that of the parent Fc protein). In addition, we recently applied this method to signaling receptors and obtained Fc-sized receptor agonists with good pharmacokinetic properties. To date, receptor agonists have been difficult to obtain in a systematic manner using conventional antibody technology. We apologize that we are unable to include these data in the current manuscript because further in-depth studies are required to validate our current findings. Furthermore, we feel that these findings are out of the scope of the current paper. Nevertheless, we are confident that significant utilities of the technology including *in vivo* data will be reported in the upcoming works.

*However, the study falls short of demonstrating the translatability of this approach. The authors also attempt to graft lasso peptides onto AAV capsid proteins, which appears to be fraught with problems. A major concern is that the overall titers of the AAV2 mutants is exceedingly low, bringing to question whether this approach can be scaled or translated. Viral transduction data using the peptide insertion mutants is primarily *in vitro* and not quantitative.*

We are puzzled why the reviewer felt that the "overall titers of the AAV2 mutants is exceedingly low". As shown in the Figure 6a (above the panels), the virus titers of the mutants are at the same level as the WT (i.e., unmodified AAV2). The absolute numbers used to describe the titer ($\sim 10^7$ vg/ μ l, or 10^{10} vg/ml) may have given the impression that they are low, compared to values like 10^{13} vg/flask that are frequently reported; however, the absolute value will differ enormously depending on way the virus preparation is formulated and how it is expressed. In our case, the numbers refer to the concentration of the non-purified virus stock obtained directly after extracting the transfected cells. Correspondingly, our titer values are in the same range as those from a typical virus production yield using a commercial kit (3.86×10^{10} vg/ml, TAKARA technical bulletin

[https://www.takarabio.com/products/gene-function/viral-transduction/adeno-associated-virus-\(aav\)/vector-systems/helper-free-expression-system-\(cmv-promoter\)?catalog=6230](https://www.takarabio.com/products/gene-function/viral-transduction/adeno-associated-virus-(aav)/vector-systems/helper-free-expression-system-(cmv-promoter)?catalog=6230)).

To clear any concerns on this matter, we provided new and stronger evidence that our approach is valid and translatable. We added new data regarding the target-specific gene transduction by new chimeric viruses containing peptide-inserted subunits in the background of wild-type subunits. As shown in the newly added

Fig.6f and 6g, the new chimeric viruses can transduce cells at low titers (MOI=1000-5000) and also exhibit receptor-specific enhancement of 2- (for PlxnB1) or 6-fold (for MET). In the case of the MET-specific virus, this number in theory translates into a potential 5-fold reduction in the virus dose required to achieve the same level of gene transduction as for wild-type. To this end, we confidently propose that this approach has a reasonable potential to be scaled and translated. Lastly, even though the data are all in vitro and no animal studies have been presented because of the nature of the current work (where the proof-of-concept study of lasso-graft is reported for the first time), quantitative data are presented in Fig.6d and e (and the new Fig.6f and g as well) along with the statistical information, which we feel sufficient.

For the chimeric VP3 insertion mutants, it is unclear how the authors determine the percentages of the different capsid components with peptide inserts in the chimeras. Capsid assembly is a stochastic process and it is likely that a largely heterogeneous population was generated. The ratios of plasmids utilized are unlikely to be actually reflected in particle composition. This is a significant flaw with the scientific premise of this arm of the study.

We agree with the reviewer that actual subunit composition of the chimeric virus cannot be accurately estimated by the ratios of the plasmid DNAs. We rephrased all the expressions regarding the subunit composition of the chimeric particles to make sure that we do not know the exact numbers, and we added the following sentence in the discussion section.

“It should be noted, however, that the chimeric virus used here could be highly heterogeneous population of mixed subunit compositions, because capsid assembly is a stochastic process. (page 9, line 329)“

Overall, this is an incremental study primarily evaluating a peptide grafting approach with the exception that the inserts are macrocyclic in nature. Technical aspects and applications of the approach appear to be cumbersome and not readily apparent, respectively.

We disagree with this conclusion as it does not reflect our understanding of the state of the field and diminishes the importance of this work as recognized by the other reviewers. Therefore, we would like to explain our understanding of the state of the field and emphasize the strength of our approach. First, peptide grafting is not a generalized methodology, even for linear peptides. There are many examples of protein detection via an inserted tag (e.g. Western blotting), but this is possible only because the binding partner (i.e. anti-tag antibody) can recognize an unstructured peptide tag. On the other hand, biologically active peptides (e.g. peptide hormones) must assume the correct 3D conformation to be recognized by their receptors, which is difficult to achieve when the conformational freedom is lost upon the insertion into a loop of a protein. Thus, straight-forward approaches for grafting medically useful peptides into a biocompatible protein scaffold have not been effective. There are notable exceptions that prove the general rule. In a previous study, a classical integrin-binding tripeptide (RGD) motif was inserted in loops of AAV capsid and the resultant mutant virus was shown to possess new binding tropism via RGD-binding integrins on cells (Girod et al Nature Genet. 1999, now included as ref 21). This is an exceptional case because the RGD motif is presented in a conformation that mimics its physiological conformation in the context of the integrin ligand fibronectin. Even in this case, the RGD motif alone cannot convey the native discrimination between integrin subtypes, resulting in a virus with a polytropic nature toward many types of cells expressing RGD-binding integrins.

It is our understanding that the field of AAV engineering has shifted its focus toward mutagenesis/selection strategies since the publication of the above study. We are familiar with many attempts to isolate AAV variants with unique and useful properties out of vast mutant populations that created either artificially or naturally (refs 22 and 24, for example). Although these attempts have yielded some promising AAV mutants, it is unfortunate that the random nature of the mutations does not allow the identification of the cellular factor(s) responsible for the improved property. In contrast, the lasso-grafting methodology introduced here starts from a peptide with defined target specificity, and we know from the beginning that the engineered virus will preferentially bind its target molecule (e.g. MET or PlxnB1). Most importantly, we can freely choose any target molecule from a pool of desired cell surface proteins, provided that we can obtain high-specificity cyclic peptides with high binding affinities via the RaPID system. Even though the 2-step procedure (first RaPID selection and then grafting) may look cumbersome on a conceptual level, we believe that the receptor design capability and much shorter development time are great advantages over existing technologies.

Reviewer #3:

Remarks to the Author:

Macrocyclic peptides are a class of synthetic compounds that are gaining increasing attention as possible therapeutic modalities, especially given their ability to target and inhibit protein-protein interactions. Despite their low molecular weights, macrocyclic peptides typically possess larger binding interfaces than traditional small molecule-based drugs, which allows for greater specificity and affinity. Furthermore, easy and rapid methods for generating macrocyclic peptides using strategies such as RaPID have emerged to make these compounds more widely and readily accessible. However, macrocyclic peptides often possess unfavorable pharmacokinetic properties that can limit their therapeutic potential. To address this issue, the authors of this paper utilize two interesting engineering strategies in which they replace the thioether ring-closure moiety of the macrocyclic peptide with a self-folding protein domain, so called "lasso grafting". In essence, they graft the pharmacophore of the macrocyclic peptide onto another protein or proteins, taking advantage of the grafted onto proteins' favorable pharmacokinetic properties while trying to maintain the binding properties of the macrocyclic peptide. In their first strategy, they take advantage of protein dimers in which the C- and N-termini of the individual proteins within the dimer are structurally held in proximity. By fusing the individual proteins into a single polypeptide sequence with the residues of the macrocyclic peptide inserted between the C- and N-termini of the parental proteins, they were able to effectively produce grafted protein dimers. They were able to utilize this strategy to graft multiple different macrocyclic peptide pharmacophores onto a uteroglobin dimer. Interestingly, while the koffs of these grafted proteins were mostly unchanged relative to the parental macrocyclic peptide, the kons where much lower. As such, the KDs for each grafted protein was in some cases substantially lower than the corresponding macrocyclic peptide. The authors' second and perhaps more powerful strategy involved grafting the pharmacophore of a macrocyclic peptide into the exposed loops of a protein. Utilizing this strategy, the authors were able to graft pharmacophores of various macrocyclic peptides easily onto many different protein scaffolds, including medically relevant antibodies. Furthermore, for proteins that possess many exposed loops, the authors were able to show that multiple pharmacophores could be grafted onto a single scaffold (ex: one in each exposed loop of an Fc domain). As such, the authors were able to show that they could produce bi-, tri-, and even tetra-specific protein binders. In an interesting application of this engineering strategy, the authors were able to graft various targeting pharmacophores onto capsid proteins of AAV particles in order to rapidly alter virus tropism. In summation, this paper presents an interesting and potentially powerful and rapid engineering strategy to 1) improve the pharmacokinetic properties of macrocyclic peptides and 2) generate multi-specific protein binders.

We thank the reviewer very much for the accurate understanding of our work and appreciating its strength.

I have only one comment to address prior to publication. Although the binding kinetics were reported for the first engineering strategy (grafting pharmacophores onto protein dimers), none were reported for the second engineering strategy (grafting pharmacophores into protein loops). It would be important to know how the binding kinetics of some of these loop-grafted proteins correlate to the parental macrocyclic peptides. It is interesting to know if the decrease in KDs reported for the dimer-grafted proteins could be rescued or even improved in the loop-grafted proteins if multiple loops within a single protein were replaced with the same target specific pharmacophore.

We thank the reviewer for raising this important point. As suggested, we performed additional experiments to analyze the binding kinetics for some of the peptide-grafted Fc proteins. The results are now incorporated in the new Fig.1c (Table) and Supplementary Fig.S2 (actual sensorgrams), and described in the text (page 5, line 143-148). The decreased affinity values in the monovalently grafted peptides were indeed rescued upon divalent presentation in the context of the Fc protein. This nicely explains why the Fc grafts were so successful in exhibiting antibody-like binding capability in both pull-down (Fig. 3) and FACS-based (Fig.4 and 5) assays.

(End of file)

REVIEWERS' COMMENTS

Reviewer #1 (Remarks to the Author):

Authors have responded satisfactorily and I support acceptance of the article.

Reviewer #2 (Remarks to the Author):

The response to reviewers from the authors and edits to the manuscript are truly appreciated. That being said, two deficiencies that render this study to a conceptual advancement rather than of paradigm shifting caliber should be noted.

1. Lasso grafting into the antibody Fc region needs to be exemplified by some functional improvement - a relevant target or therapeutic cargo demonstrating feasibility in vivo (e.g., *Sci Transl Med.* 2020. May 27;12(545).e1163/1359.). Without such, it is unclear how the lasso-grafted peptides constitute an improvement over existing linear peptide or fusion approaches being evaluated in the clinic.

2. A major hurdle with "targeted" AAV vectors is the inability to avoid liver sequestration and reach the target in functionally appreciable quantities/doses. While this reviewer appreciates the lasso grafting approach onto a viral surface, applications in AAV gene therapy should be exemplified in vivo similar to other peptide insertion strategies (Li and Samulski, 2020, *Nat Reviews*; Grimm et al., 2020, *Nat Commun*; Gradinaru et al., *Nat Biotech*, 2016; *Nat Neurosci*, 2017; *Nat Methods*, 2019; *Nat Neurosci*, 2020). Without these demonstrations of functional improvement, the current study is primarily an engineering approach with biochemical/biophysical characterization.

Please note that it is not the intent of this reviewer to diminish the importance of this contribution. The lasso grafting approach will add to the arsenal of protein display approaches. However, the novelty and likelihood of influencing thinking in the field are low. This reviewer will defer to the editors in deciding the overall impact of the study and suitability for publication.